# PowerSGD: Practical Low-Rank Gradient Compression for Distributed Optimization

**Thijs Vogels**
EPFL
Lausanne, Switzerland
thijs.vogels@epfl.ch

**Sai Praneeth Karimireddy**
EPFL
Lausanne, Switzerland
sai.karimrieddy@epfl.ch

**Martin Jaggi**
EPFL
Lausanne, Switzerland
martin.jaggi@epfl.ch

## Abstract

We study lossy gradient compression methods to alleviate the communication bottleneck in data-parallel distributed optimization. Despite the significant attention received, current compression schemes either do not scale well, or fail to achieve the target test accuracy. We propose a new low-rank gradient compressor based on power iteration that can i) compress gradients rapidly, ii) efficiently aggregate the compressed gradients using all-reduce, and iii) achieve test performance on par with SGD. The proposed algorithm is the only method evaluated that achieves consistent wall-clock speedups when benchmarked against regular SGD using highly optimized off-the-shelf tools for distributed communication. We demonstrate reduced training times for convolutional networks as well as LSTMs on common datasets. Our code is available at https://github.com/epfml/powersgd.

## 1  Introduction

Synchronous data-parallel SGD is the most common method for accelerating training of deep learning models (Dean et al., 2012; Iandola et al., 2015; Goyal et al., 2017). Because the gradient vectors of such models can be large, the time required to share those gradients across workers limits the scalability of deep learning training (Seide et al., 2014; Iandola et al., 2015; Lin et al., 2018).

Previous work proposes lossy gradient compression as a solution to this issue. Notable examples include replacing the coordinates of the gradient with only their sign (Seide et al., 2014; Carlson et al., 2015; Bernstein et al., 2018, 2019; Karimireddy et al., 2019), quantizing the individual coordinates (Alistarh et al., 2017; Wen et al., 2017), and low-rank approximation of the gradient (Wang et al., 2018). While these works demonstrate speedups over full-precision SGD in some settings, we find that their speedups vanish with a fast network and highly optimized communication backend, even on commodity hardware. Some prior work also suffers from degraded test accuracy compared to SGD. We combine three observations to fix these issues: i) Linear compressor operators achieve scalability by enabling aggregation using all-reduce. ii) Error feedback ensures convergence with general biased compressors. iii) Low-rank updates enable aggressive compression without sacrificing quality.

First, we explore the properties of various gradient compression schemes for SGD and identify which ones are crucial for high scalability. In particular, we note that currently proposed gradient compressors are not linear. Their compressed messages cannot be added up hierarchically, unlike raw gradients. This prevents current compressed SGD algorithms from aggregating gradients using an efficient *reduce* operation and instead require a *gather* operation. Current deep learning frameworks rely either solely or predominantly on all-reduce, which is key to why regular SGD scales well with fast communication hardware (cf. Awan et al., 2018; Panda et al., 2019).

Secondly, it was recently shown that using error feedback (i.e. storing the difference between the computed and compressed gradient, and reinserting it at the next iteration) improves both convergence

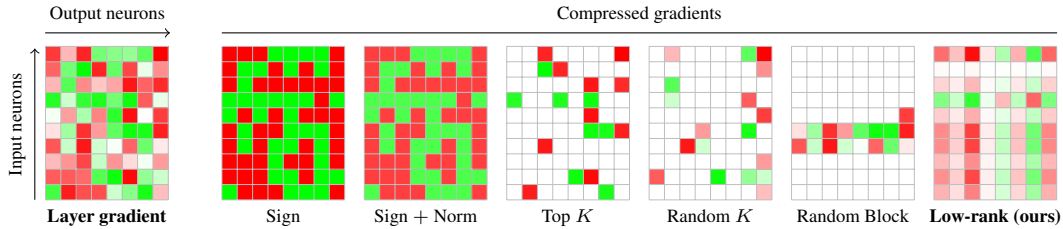

Figure 1: Compression schemes compared in this paper. Left: Interpretation of a layer's gradient as a matrix. Right: The output of various compression schemes. Implementation details in Appendix G.

and generalization for compression schemes (Karimireddy et al., 2019). This can enable general biased gradient compression schemes to reach the target test accuracy.

Thirdly, there is growing evidence that the generalization ability of modern over-parameterized deep learning models is related to low-rankedness (Arora et al., 2018; Martin & Mahoney, 2018; Collins et al., 2018). Using a low-rank update (as we do) can be viewed as implicitly performing spectral regularization (Gunasekar et al., 2018) and hence can be expected to have good generalization properties (Yoshida & Miyato, 2017). Further, Wang et al. (2018) show that the eigenspectrum of the stochastic gradients for deep learning models decays, suggesting that a rank-based schemes can get away with aggressive compression without sacrificing convergence.

In this work, we design POWERSGD with the above observations in mind. POWERSGD computes a low-rank approximation of the gradient using a generalized *power* iteration (known as subspace iteration (Stewart & Miller, 1975)). The approximation is computationally light-weight, avoiding any prohibitively expensive Singular Value Decomposition. To improve the quality of the efficient approximation, we *warm-start* the power iteration by reusing the approximation from the previous optimization step. Using all-reduce gradient aggregation, we empirically demonstrate that POWERSGD achieves wall-clock speedups over regular SGD in a 16-GPU setting, even with the optimized NCCL communication backend on a fast network (and is the only algorithm to do so.) By compressing gradients more than $120\times$, we reduce communication time (including coding and decoding) by $54\%$ for RESNET18 on CIFAR10 and by $90\%$ for an LSTM on WIKITEXT-2. End-to-end wall-clock training time to full test quality is reduced by $24\%$ for RESNET18 and by $55\%$ for the LSTM.

## 2 Related work

**Gradient compression** A variety of compression schemes (Figure 1) have been proposed: Alistarh et al. (2017) and Wen et al. (2017) quantize each gradient coordinate; Seide et al. (2014); Carlson et al. (2015); Bernstein et al. (2018, 2019) and Karimireddy et al. (2019) replace each coordinate of the gradient with its sign; Lin et al. (2018); Stich et al. (2018) and Wangni et al. (2018) use the largest few coordinates; and Konečný et al. (2016) and Wang et al. (2018) use a low-rank approximation.

Spectral Atomo by Wang et al. (2018) is perhaps the closest to our work. It performs importance sampling of the gradient's singular vectors and is an unbiased compression scheme. It requires, however, a full Singular Value Decomposition every iteration and is hence computationally impractical.

**Commutative compression and addition** Yu et al. (2018) stress that commutability of compression with gradient addition enables efficient aggregation with *ring all-reduce*. Most compressors, however, lack this property. Yu et al. utilize temporally-consistent correlations between gradients coordinates to compress them linearly. POWERSGD has a similar property that we call 'linearity'.

**Error feedback** First introduced in (Seide et al., 2014) and analyzed in (Stich et al., 2018) for the convex case, error feedback involves computing the difference between a worker's gradient and the compressed gradient (i.e. *error*) and adding it back to the next gradient (*feedback*). Karimireddy et al. (2019) and Stich & Karimireddy (2019) further develop and generalize the framework of error feedback with improved rates. In the non-convex setting, Karimireddy et al. (2019) show that error feedback is crucial both for convergence and generalization when using biased compressors (e.g. sign or top-$K$). In general, biased compression schemes equipped with error feedback tend to out-perform their unbiased counterparts. The practical algorithm by Lin et al. (2018) is also as an approximate top-$K$ compressor with error feedback.

**Low-rank methods**  Recent works argue that in modern over-parameterized deep networks, the final model learnt has a 'low stable rank' (Martin & Mahoney, 2018; Li et al., 2018). This can partially explain their impressive generalization properties despite being substantially overparameterized (Arora et al., 2018). Adding explicit spectral regularization has shown to further improve the performance of such models (Mazumder et al., 2010; Yoshida & Miyato, 2017). Using a low-rank update (as we do) can be viewed as implicitly performing a similar regularization (Gunasekar et al., 2018). If the target matrices are known to be exactly low-ranked (instead of just low stable rank), Yurtsever et al. (2017) show that it is sometimes possible to converge to the optima using low rank approximations of the gradients without the need for error feedback.

## 3   Method

In data-parallel optimization of machine learning models, a number of $W$ workers share the same model parameters $\mathbf{x} \in \mathbb{R}^d$. They iteratively update $\mathbf{x}$ by computing independent stochastic gradients, aggregating these gradients by averaging[1], and updating the model parameters based on this aggregate.

---

**Algorithm 1** Rank-$r$ POWERSGD compression

---

1: The update vector $\Delta_w$ is treated as a list of tensors corresponding to individual model parameters. Vector-shaped parameters (biases) are aggregated uncompressed. Other parameters are reshaped into matrices. The functions below operate on such matrices independently. For each matrix $M \in \mathbb{R}^{n \times m}$, a corresponding $Q \in \mathbb{R}^{m \times r}$ is initialized from an i.i.d. standard normal distribution.
2: **function** COMPRESS+AGGREGATE(update matrix $M \in \mathbb{R}^{n \times m}$, previous $Q \in \mathbb{R}^{m \times r}$)
3:     $P \leftarrow MQ$
4:     $P \leftarrow$ ALL REDUCE MEAN$(P)$                     ▷ Now, $P = \frac{1}{W}(M_1 + \ldots + M_W)Q$
5:     $\hat{P} \leftarrow$ ORTHOGONALIZE$(P)$                          ▷ Orthonormal columns
6:     $Q \leftarrow M^{\top}\hat{P}$
7:     $Q \leftarrow$ ALL REDUCE MEAN$(Q)$                     ▷ Now, $Q = \frac{1}{W}(M_1 + \ldots + M_W)^{\top}\hat{P}$
8:     **return** the compressed representation $(\hat{P}, Q)$.
9: **end function**
10: **function** DECOMPRESS$(\hat{P} \in \mathbb{R}^{n \times r}, Q \in \mathbb{R}^{m \times r})$
11:     **return** $\hat{P}Q^{\top}$
12: **end function**

---

**POWERSGD compression**  We approximate each layer in the model independently. The parameters of fully-connected layers (dense matrix multiplication) and their gradients have an inherent matrix structure. The parameters of convolutional layers can be naturally interpreted as fully-connected layers applied repeatedly over a 2D grid of inputs. Practically, this amounts to flattening input and kernel dimensions in the 4D gradient tensors. Neural networks also contain bias vectors, but these typically constitute a tiny fraction of the parameter space and can be aggregated uncompressed.

For each parameter's gradient $M \in \mathbb{R}^{n \times m}$, the aim of rank-$r$ matrix approximation is to find matrices $P \in \mathbb{R}^{n \times r}$ and $Q \in \mathbb{R}^{m \times r}$ such that $PQ^{\top}$ approximates $M$ well. POWERSGD uses a single step of subspace iteration—*power* iteration generalized to $r > 1$—to compute such an approximation. This involves performing one right multiplication, one left multiplication, and an orthogonalization. We use the Gram-Schmidt procedure to orthogonalize our matrices since they have very few columns (1–4), and this is the most expensive part of the compression procedure. Further, we 'warm-start' the subspace iteration by reusing the approximation computed at the previous step. With the inclusion of warm-start, a *single* step of subspace iteration yields a factorization $M \sim PQ^{\top}$ with the same performance as the best rank-$r$ approximation from an expensive Singular Value Decomposition.

**Efficient aggregation between workers**  In data-parallel optimization, we want to approximate the *average* of the worker's gradients. Suppose POWERSGD operates on a list of corresponding gradients $[M_1 \ldots M_W]$ from $W$ workers. Both occurrences of $M$ in the algorithm are a (linear) matrix multiplication followed by a (linear) mean reduction over workers. This introduces a practical invariance: execution on 1 worker with batch size $B \times W$ is equivalent to execution on $W$ workers with batch size $B$ each. We call this property 'linearity'. Refer to Appendix A.3 for more details.

An important benefit of the POWERSGD's linearity is that it can be implemented using the **all-reduce** protocol as opposed to needing a gather operation. To illustrate the difference, suppose that we want to compute the sum of $W$ matrices $\sum_{i=1}^{W} M_i$ for $W = 4$. The all-reduce method can use associativity of addition to rewrite the computation as $(M_1 + M_2) + (M_3 + M_4)$.

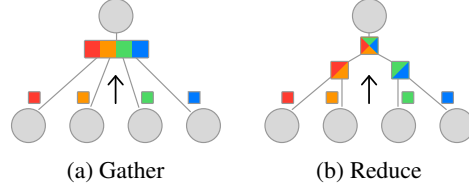

(a) Gather        (b) Reduce

This enables a divide-and-conquer approach and allows the summation task to be split over multiple workers, as illustrated on the right. With $W$ workers, both the computation and the communication time scale as $\mathcal{O}(\log W)$ for all-reduce, compared to $\mathcal{O}(W)$ for all-gather.

In addition to improved scaling, all-reduce communication is preferred over a parameter-server setting because it avoids *double compression*. With a parameter server, both the 'clients → server' and 'server → clients' communication have to be compressed (Caldas et al., 2018; Bernstein et al., 2019; Seide et al., 2014). We avoid this by merging compression and aggregation into one step.

**Error-feedback SGD**    Since the POWERSGD scheme is biased (i.e. compressing and decompressing a random gradient does not yield the original in expectation), we use error feedback (Seide et al., 2014; Karimireddy et al., 2019). Our version of error feedback (Algorithm 2) extends the original by introducing post-compression *momentum*. This simple extension allows us to reuse the same learning rate and hyper-parameters as those tuned for SGD with momentum.

---

**Algorithm 2** Distributed Error-feedback SGD with Momentum

---

1: **hyperparameters:** learning rate $\gamma$, momentum parameter $\lambda$
2: **initialize** model parameters $\mathbf{x} \in \mathbb{R}^d$, momentum $\mathbf{m} \leftarrow \mathbf{0} \in \mathbb{R}^d$, replicated across workers
3: **at** each worker $w = 1, \ldots, W$ **do**
4:      **initialize** memory $\mathbf{e}_w \leftarrow \mathbf{0} \in \mathbb{R}^d$
5:      **for** each iterate $t = 0, \ldots$ **do**
6:          Compute a stochastic gradient $\mathbf{g}_w \in \mathbb{R}^d$.
7:          $\Delta_w \quad\leftarrow \mathbf{g}_w + \mathbf{e}_w$                          ▷ Incorporate error-feedback into update
8:          $\mathcal{C}(\Delta_w) \leftarrow \text{COMPRESS}(\Delta_w)$
9:          $\mathbf{e}_w \quad\leftarrow \Delta_w - \text{DECOMPRESS}(\mathcal{C}(\Delta_w))$               ▷ Memorize local errors
10:         $\mathcal{C}(\Delta) \quad\leftarrow \text{AGGREGATE}(\mathcal{C}(\Delta_1), \ldots, \mathcal{C}(\Delta_W))$            ▷ Exchange gradients
11:         $\Delta' \quad\leftarrow \text{DECOMPRESS}(\mathcal{C}(\Delta))$               ▷ Reconstruct an update $\in \mathbb{R}^d$
12:         $\mathbf{m} \quad\leftarrow \lambda \mathbf{m} + \Delta'$
13:         $\mathbf{x} \quad\leftarrow \mathbf{x} - \gamma (\Delta' + \mathbf{m})$
14:      **end for**
15: **end at**

---

## 4   Analysis of POWERSGD

In this section, we consider different aspects of POWERSGD in isolation and hope to empirically understand: i) the effect of using error feedback, ii) the effect of 'warm-start', and iii) the trade-off between test accuracy and compression rate with varying approximation rank.

### 4.1   Effect of error feedback

Using error-feedback SGD as a base algorithm for POWERSGD has two advantages. First, it enables our use of a biased compressor. Secondly, EF-SGD improves convergence and obtains better test accuracy (Karimireddy et al., 2019).

To illustrate the improved test accuracy, we compare POWERSGD—a biased compressor with error feedback—against an unbiased low-rank approximation. To approximate a matrix $M \in \mathbb{R}^{n \times m}$, the unbiased rank-$r$ approximator samples a random matrix $U \in \mathbb{R}^{m \times r}$ such that $\mathbb{E}[UU^\top] = I_m$ and outputs $(MU, U)$ as the low-rank approximation. This scheme is unbiased since

$$\mathbb{E}[(MU)U^\top] = M \, \mathbb{E}[UU^\top] = MI = M \,.$$

POWERSGD is the natural biased counterpart of this unbiased scheme. Table 1 demonstrates that our biased approximator with error feedback outperforms the unbiased operator on image classification.

Table 1: Rank-based compression with and without error feedback. The biased POWERSGD outperforms an unbiased linear rank-$r$ compressor on test accuracy.

| Algorithm | Test accuracy | Data/epoch |
|---|---|---|
| SGD | 94.3% | 1023 MB |
| Rank-1 POWERSGD | 93.6% | 4 MB |
| Rank-2 POWERSGD | 94.4% | 8 MB |
| Unbiased Rank 1 | 71.2% | 3 MB |
| Unbiased Rank 2 | 75.9% | 4 MB |

Table 2: Best rank-2 approximation vs. POWERSGD. Warm-start improves test accuracy, even matching the performance of the best rank-2 approximation.

| Algorithm | Test accuracy |
|---|---|
| Best approximation | 94.4% |
| Warm start (default) | 94.4% |
| Without warm start | 94.0% |

Table 3: POWERSGD with varying rank. With sufficient rank, POWERSGD accelerates training of a RESNET18 and an LSTM by reducing communication, achieving test quality on par with regular SGD in the same number of iterations. The time per batch includes the forward/backward pass (constant). See Section 5 for the experimental setup.

**Image classification — RESNET18 on CIFAR10**

| Algorithm | Test accuracy | Data sent per epoch | | Time per batch | |
|---|---|---|---|---|---|
| SGD | 94.3% | 1023 MB | $(1\times)$ | 312 ms | $+0\%$ |
| Rank 1 | 93.6% | 4 MB | $(243\times)$ | 229 ms | $-26\%$ |
| Rank 2 | 94.4% | 8 MB | $(136\times)$ | 239 ms | $-23\%$ |
| Rank 4 | 94.5% | 14 MB | $(72\times)$ | 260 ms | $-16\%$ |

**Language modeling — LSTM on WIKITEXT-2**

| Algorithm | Test perplexity | Data sent per epoch | | Time per batch | |
|---|---|---|---|---|---|
| SGD | 91 | 7730 MB | $(1\times)$ | 300 ms | $+0\%$ |
| Rank 1 | 102 | 25 MB | $(310\times)$ | 131 ms | $-56\%$ |
| Rank 2 | 93 | 38 MB | $(203\times)$ | 141 ms | $-53\%$ |
| Rank 4 | 91 | 64 MB | $(120\times)$ | 134 ms | $-55\%$ |

## 4.2 Effect of warm-start

POWERSGD does not compute the best rank-$r$ approximation of a gradient matrix, but uses a cheaper, low-fidelity approximation based on power iteration. Comparing the time per batch of POWERSGD and Spectral Atomo in Table 6, we see the importance of avoiding a Singular Value Decomposition. With gradients shaped as in POWERSGD, computing the SVD of a stochastic gradient takes 673ms, the equivalent of computing 6 mini-batch gradients. In contrast, one full step of rank-2 POWERSGD, including communication between 16 workers, takes only 105ms.

Given that we only use a single step of power iteration, the quality of the approximation suffers—compare the test accuracy of 'without warm start' and 'best approximation' in Table 2. A key feature of POWERSGD is the *warm start* strategy which reuses previously computed matrix approximations to initialize the power iteration algorithm. If the matrix on which we perform power iteration remains constant, then this recovers the best rank-$r$ approximation (see Theorem I in the Appendix). We argue that this strategy sometimes makes sense even if the underlying matrices are varying.

Suppose we approximate the sequence of gradient matrices $\{M_t\}$ at timesteps $t$. At timestep $t$, we leverage the previous factorization $M_{t-1} \approx P_{t-1}Q_{t-1}^\top$. If $M_t \approx M_{t-1}$ then we would benefit from reusing $P_{t-1}$ and $Q_{t-1}$ as our starting point. While this is unlikely to be true, if $M_t$ and $M_{t-1}$ are stochastic approximations of the full gradient, we can expect that $\mathbb{E}[M_t] \approx \mathbb{E}[M_{t-1}]$ since the function is smooth and we only take small update steps. The result is akin to Oja's algorithm for *stochastic power iteration* (Oja, 1982), and hence could result in an improved approximation quality. As we show empirically in Table 2, this 'warm starting' strategy is sufficient to close the gap in test accuracy between POWERSGD and the much more expensive best rank-$r$ approximation.

## 4.3 Effect of varying the rank

POWERSGD allows users to choose the rank of its gradient approximations. The trade-off between approximation quality and compression, decompression and transfer cost is explored in Table 3. In both the image classification and language modeling tasks we explore, the test quality achieved by POWERSGD grows with increasing rank. In both cases, it reaches a quality that is as good, or even slightly better than regular SGD.

Table 4: Comparing different compression operators for Error-feedback SGD in a unified setting; running 300 epochs of Error-feedback SGD with Momentum (Algorithm 2) with a learning rate tuned for full-precision SGD on 16 GPUs for CIFAR10. Note that the variations of POWERSGD with ranks 2 and 7 strikes the best balance between the achieved test accuracy and time per batch (total time for forward, backward, compression, decompression, and gradient aggregation).

| | | Test accuracy | Sent/epoch | All-reduce | Time/batch |
|---|---|---|---|---|---|
| No compression | | 94.3% | 1023 MB | ✓ | 312 ms |
| Medium | **Rank 7** | 94.6% | 24 MB | ✓ | 285 ms |
| | Random Block | 93.3% | 24 MB | ✓ | 243 ms |
| | Random K | 94.0% | 24 MB | ✓ | 540 ms |
| | Sign+Norm | 93.9% | 32 MB | ✗ | 429 ms |
| | Top K | 94.4% | 32 MB | ✗ | 444 ms |
| High | **Rank 2** | 94.4% | 8 MB | ✓ | 239 ms |
| | Random Block | 87.8% | 8 MB | ✓ | 240 ms |
| | Random K | 92.6% | 8 MB | ✓ | 534 ms |
| | Top K | 93.6% | 8 MB | ✗ | 411 ms |

# 5 Results

This section demonstrates the practicality of POWERSGD for distributed optimization of deep neural networks. We show that the compression scheme of POWERSGD i) is fast and matches test performance of SGD, ii) scales well with increasing workers even with a sub-optimal communication backend, and iii) significantly reduces training time for larger models.

Most of the analysis is performed on CIFAR10, in the setting described in the table on the right. We verify the generality of POWERSGD by an additional evaluation of an LSTM for language modeling on WIKITEXT-2. We use 16 GPUs on 8 machines, connected through a fast (10Gbit/s) network. To obtain meaningful timings, we have aimed to optimize all compared optimizers to a similar level. We provide a list of our performance optimizations in Appendix H. Throughout these results, we tune the learning rate for full-precision SGD, and use the *same* parameters for POWERSGD and other compression algorithms that use error feedback with momentum. Learning rates for the compared-to Spectral Atomo (Wang et al., 2018) and Signum (Bernstein et al., 2019) were separately tuned cf. Appendix I.

| Default experimental setting | |
|---|---|
| Dataset | CIFAR10 |
| Architecture | RESNET18 |
| Number of workers | 16 |
| Backend | NCCL (fastest in PYTORCH) |
| Batch size | $128 \times$ number of workers |
| Momentum | 0.9 |
| Learning rate | Tuned for 16 workers — $0.1 \times 16$ for SGD. Scaled linearly by the number of workers |
| LR decay | $/10$ at epoch 150 and 250 |
| LR warmup | Linearly within 5 epochs, starting from the single-worker LR |
| # Epochs | 300 |
| Weight decay | $10^{-4}$, 0 for BatchNorm parameters |
| Repetitions | 3, with varying seeds |
| Error bars | min — max |

## 5.1 Comparison with other compressors

Error feedback in compressed optimization enables the use of a multitude of compression schemes, including biased ones. The potential compression operators illustrated in Figure 1 are compared in Table 4. We evaluate compressors based on the test accuracy achieved and the total time taken to process one mini-batch. The former is a holistic measure of the accuracy of the compression operator, and the latter is the net time required for a forward pass, backward pass, gradient compression and decompression and gradient communication. We study two compression regimes—medium and high.

At around $32\times$ compression, achieved by sign-based methods, all compression schemes (other than Random Block) achieve test accuracy close to full-precision SGD. This implies that all schemes in this regime (other than Random Block) obtain a good-enough compression quality. At high compression ($128\times$), POWERSGD particularly stands out as the only method to achieve the target test accuracy.

In both the medium and high compression settings, the only schemes to be faster than full-precision SGD are POWERSGD and Random Block. Note that both are simple linear schemes and hence support all-reduce. While Random $K$ also supports all-reduce, the overhead for random memory access during both the compression and decompression stages is substantial, making it slower overall

Table 5: Breakdown of time spent (in seconds) in one iteration of RESNET18 training. Because POWERSGD (Rank 2) uses all-reduce, time spent encoding/decoding gradients is constant.
■ Forward pass, ■ Backward pass, ■ Gradient exchange, ▨ Encoding and decoding.

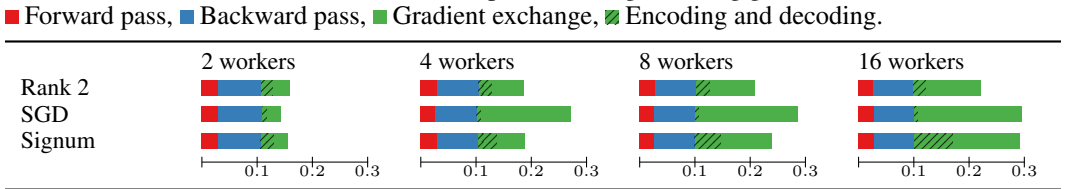

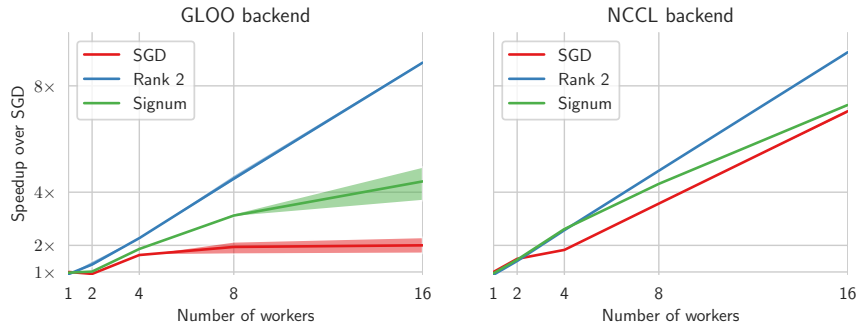

Figure 3: Scaling of POWERSGD on CIFAR10 compared to full-precision SGD and Signum (Bernstein et al., 2019) on two communication backends. The batch size increases linearly with the number of workers. We compare training time for one epoch to 1-worker SGD. Note that the faster NCCL backend used throughout benefits the baselines more than our method.

than SGD. Thus, on modern GPU-enabled infrastructure, POWERSGD, which relies on matrix multiplication, is faster and much more accurate than the other compression schemes.

## 5.2   Scalability of POWERSGD

Here we investigate how POWERSGD scales with an increasing number of workers, shedding light on what we can expect if we use a significantly larger number of workers. Additionally, we investigate how these results depend on the choice of communication backend. We benchmark POWERSGD against SGD and Signum (signSGD with majority vote) from Bernstein et al. (2019) which we believe is the current state-of-the-art for distributed algorithms.

Table 5 provides a detailed breakdown of the time spent for each mini-batch (i.e. one step) into the forward pass, backward pass, gradient exchange (communication), and compression/decompression. The time spent in the forward and backward pass is constant across all algorithms and numbers of workers. Since both SGD and POWERSGD use all-reduce, the gradient communication time (solid green in Table 5) scales gracefully with increasing number of workers. Signum—which uses all-gather instead of all-reduce—has a steeper increase. It has comparable time to POWERSGD for 4 workers but becomes more expensive for 16 workers.

There is another, more subtle, consequence of all-reduce vs. all-gather on the decoding times. In all-reduce, the *aggregation* step and the *communication* step happen simultaneously. Each worker receives a pre-aggregated gradient, making the cost of decompression independent of the number of workers. On the other hand, in all-gather, a worker receives $W$ compressed gradients that need to be individually decompressed and aggregated (either using majority vote or averaging). The time for decompression with all-gather therefore scales linearly with number of workers. This shows when comparing the hatcheted regions in Table 5. This observation speaks to the importance of the reduce operation for scalability.

We next study two different backends—the more optimized NCCL and the slower GLOO. All three methods scale reasonably well with the optimized NCCL backend, although Signum has a slope less than 1 in the log-log plot, indicating sub-linear scaling. On the slower GLOO backend, POWERSGD is notably the only method that retains excellent scaling due to its high compression rate.

Table 6: Results on CIFAR10. Contrary to rank-2 Spectral Atomo (Wang et al., 2018) and Signum (Bernstein et al., 2019), POWERSGD achieves the same test accuracy as full-precision SGD within the default epoch budget.

| Algorithm | Test accuracy | Data/epoch | Time per batch | |
|---|---|---|---|---|
| SGD | 94.3% | 1023 MB | 312 ms | +0% |
| Atomo | 92.6% | 113 MB | 948 ms | +204% |
| Signum | 93.6% | 32 MB | 301 ms | −3% |
| **Rank 2** | 94.4% | 8 MB | 239 ms | −23% |

Table 7: In **language modeling**, rank-4 POWERSGD achieves the target test accuracy and provides a significant speedup over SGD.

| Algorithm | Test perplexity | Data/epoch | Time per batch | |
|---|---|---|---|---|
| SGD | 91 | 7730 MB | 300 ms | +0% |
| Signum | 142 | 242 MB | 424 ms | +41% |
| **Rank 4** | 91 | 64 MB | 134 ms | −55% |

### 5.3 Other tasks and methods

In Table 6, we compare POWERSGD against the state-of-the-art compressed optimization algorithms Signum and Spectral Atomo. The cost of performing a full SVD at each step renders Spectral Atomo impractical in a high-performance setting, especially considering that it fails to match the test accuracies of the other methods. Signum performs much better, proving a minor speedup over SGD. POWERSGD is the fastest and most accurate of the compared methods.

The advantage of POWERSGD truly shows when using really large models, i.e. where the communication actually becomes a bottleneck. To verify this, we run Signum, full-precision SGD, and POWERSGD to train an LSTM on a language modeling task which has a substantially larger model size than RESNET18 (see Appendix F). To match the test score of full-precision SGD, we needed to use a rank-4 approximation (see Section 4.3). POWERSGD reduces communication by 90% and the overall running time by 55%, while Signum becomes slower than full-precision SGD and also obtains a worse test score.

Convergence curves on test accuracy corresponding to Tables 3, 6 and 7 are provided in Appendix C. In those figures, you can read our improvements in time-to-accuracy for any target accuracy. We also provide a case study on using PowerSGD for a novel task (language modeling with transformers on WIKITEXT-2) and more workers (32) on the public cloud in Appendix D.

## 6 Conclusion

Gradient compression is a promising approach to tackling the communication bottleneck in synchronous distributed optimization. Thus far, however, it has not found widespread adoption because existing compression schemes either run slower than SGD with optimized all-reduce gradient aggregation, or more importantly do not reach the same test performance. We see POWERSGD as the first practical gradient compression method, and believe it is ready for adaptation in practice.

The key to the practicality of POWERSGD is its linear compression scheme that is cheap to compute and allows for all-reduce gradient aggregation, while simultaneously matching the test performance of full-precision SGD. This speedup gained over SGD actually *increases* for larger models such as those commonly found in NLP. Further, as a result of our modifications to the error feedback algorithm, POWERSGD is a plug-in replacement for SGD with momentum, avoiding the need for additional hyper-parameter tuning. We expect that these properties of POWERSGD will enable training of even larger models with even more workers than what is possible with full-precision SGD.

While POWERSGD enables faster training with larger batch sizes, increasing batch sizes are known to eventually suffer from a 'generalization gap' (Shallue et al., 2018). This is an orthogonal issue that we see as the next step towards solving large-scale training. In our experiments, we have observed that POWERSGD can achieve higher test accuracy than SGD. Combined with the intriguing links between low-rankedness and generalization, this indicates that POWERSGD may also be helpful for closing the generalization gap in large batch training.

**Acknowledgements**

We thank Alp Yurtsever and Tao Lin for valuable discussions and the reviewers for their feedback. This project was supported by SNSF grant 200021_175796, as well as a Google Focused Research Award.

## Footnotes

[1]Bernstein et al. (2019) propose Signum which aggregates 1-bit gradients by majority voting instead of averaging.

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
