[Supplementary Material]

# PowerSGD: Practical Low-Rank Gradient Compression for Distributed Optimization

**Thijs Vogels**
EPFL
Lausanne, Switzerland
`thijs.vogels@epfl.ch`

**Sai Praneeth Karimireddy**
EPFL
Lausanne, Switzerland
`sai.karimrieddy@epfl.ch`

**Martin Jaggi**
EPFL
Lausanne, Switzerland
`martin.jaggi@epfl.ch`

## Abstract

We study lossy gradient compression methods to alleviate the communication bottleneck in data-parallel distributed optimization. Despite the significant attention received, current compression schemes either do not scale well, or fail to achieve the target test accuracy. We propose a new low-rank gradient compressor based on power iteration that can i) compress gradients rapidly, ii) efficiently aggregate the compressed gradients using all-reduce, and iii) achieve test performance on par with SGD. The proposed algorithm is the only method evaluated that achieves consistent wall-clock speedups when benchmarked against regular SGD using highly optimized off-the-shelf tools for distributed communication. We demonstrate reduced training times for convolutional networks as well as LSTMs on common datasets. Our code is available at `https://github.com/epfml/powersgd`.

## 1 Introduction

Synchronous data-parallel SGD is the most common method for accelerating training of deep learning models (Dean et al., 2012; Iandola et al., 2015; Goyal et al., 2017). Because the gradient vectors of such models can be large, the time required to share those gradients across workers limits the scalability of deep learning training (Seide et al., 2014; Iandola et al., 2015; Lin et al., 2018).

Previous work proposes lossy gradient compression as a solution to this issue. Notable examples include replacing the coordinates of the gradient with only their sign (Seide et al., 2014; Carlson et al., 2015; Bernstein et al., 2018, 2019; Karimireddy et al., 2019), quantizing the individual coordinates (Alistarh et al., 2017; Wen et al., 2017), and low-rank approximation of the gradient (Wang et al., 2018). While these works demonstrate speedups over full-precision SGD in some settings, we find that their speedups vanish with a fast network and highly optimized communication backend, even on commodity hardware. Some prior work also suffers from degraded test accuracy compared to SGD. We combine three observations to fix these issues: i) Linear compressor operators achieve scalability by enabling aggregation using all-reduce. ii) Error feedback ensures convergence with general biased compressors. iii) Low-rank updates enable aggressive compression without sacrificing quality.

First, we explore the properties of various gradient compression schemes for SGD and identify which ones are crucial for high scalability. In particular, we note that currently proposed gradient compressors are not linear. Their compressed messages cannot be added up hierarchically, unlike raw gradients. This prevents current compressed SGD algorithms from aggregating gradients using an efficient *reduce* operation and instead require a *gather* operation. Current deep learning frameworks rely either solely or predominantly on all-reduce, which is key to why regular SGD scales well with fast communication hardware (cf. Awan et al., 2018; Panda et al., 2019).

Secondly, it was recently shown that using error feedback (i.e. storing the difference between the computed and compressed gradient, and reinserting it at the next iteration) improves both convergence

Figure 1: Compression schemes compared in this paper. Left: Interpretation of a layer's gradient as a matrix. Coordinate values are color coded (**positive**, **negative**). Right: The output of various compression schemeson the same input. Implementation details arein

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

[2]`https://github.com/pytorch/fairseq/tree/920b85d4bd39e181229db5639c701c854c83ec5c/` `examples/language_model`

[3] 'reduce'+'gather' (parameter server communication) with GLOO takes longer than all-gather with NCCL, as shown in Appendix B. NCCL in PYTORCH currently lacks support for a 'gather' operator.

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

# Appendix

## A   Discussion of convergence

The proof of convergence of EF-SGD with momentum can be derived by incoporating a few key changes to the proof of Karimireddy et al. (2019): i) we are in a multi-worker setting, and ii) we incorporate the techniques introduced by Ghadimi & Lan (2016) to handle the additional *momentum*. Further, $\|\cdot\|^2$ unless otherwise specified is always the standard euclidean norm for vectors, and is the *Frobenius* norm for matrices.

Suppose that we want to minimize a continuous (possibly) non-convex function $f \colon \mathbb{R}^d \to \mathbb{R}$:

$$f^\star = \min_{\mathbf{x} \in \mathbb{R}^d} f(\mathbf{x}) \,.$$

The classic stochastic gradient algorithm (SGD) Robbins & Monro (1951) when adapted to the distributed optimization setting performs iterations of the form

$$\mathbf{x}_{t+1} := \mathbf{x}_t - \gamma \, \mathbf{g}_t \,, \text{ where} \tag{1}$$

$$\mathbf{g}_t = \frac{1}{W} \sum_{w=1}^{W} \mathbf{g}_{t,w} \qquad \text{and} \qquad \mathbb{E}[\mathbf{g}_t] = \nabla f(\mathbf{x}_t) \,.$$

Here $\gamma \in \mathbb{R}$ is the step-size (or learning-rate) and $\mathbf{g}_{t,w}$ is the stochastic gradient computed by the $w$th worker for $w \in \{1, \ldots, W\}$ workers.

Now EF-SGD (Algorithm 2) when run on the $W$ workers with step-size $\gamma$ and momentum parameter $\lambda$ can be rewritten making the dependence on iteration $t$ explicit as follows:

$$
\begin{aligned}
\Delta'_t &= \text{DECOMPRESS}(\text{COMPRESS}(\mathbf{g}_t + \mathbf{e}_t)) \,, \\
\mathbf{m}_{t+1} &= \Delta'_t + \lambda \mathbf{m}_t \,, \\
\mathbf{x}_{t+1} &= \mathbf{x}_t - \gamma(\Delta'_t + \mathbf{m}_{t+1}) \,, \text{ and} \\
\mathbf{e}_{t+1} &= (\mathbf{g}_t + \mathbf{e}_t) - \Delta'_t \,.
\end{aligned}
\tag{2}
$$

### A.1   Eigen compression

**Assumption A** (Eigen compression). *Consider any matrix $M = g_t + e_t$ encountered during the run of Algorithm 2 such that $M$ is of rank $R$. Further, suppose that $\mathcal{C}_r(M)$ is the best rank-$r$ approximation of $M$ i.e.*

$$\mathcal{C}_r(M) = \arg\min_C \|M - C\|^2 \,.$$

*Then we assume that there exists a $\delta_{e,r} > 0$ such that*

$$\|M - \mathcal{C}_r(M)\|^2 \le (1 - \delta_{e,r}) \|M\|^2 \ \ a.s.$$

We state the below standard fact from linear algebra.

**Remark 1** (Best rank-$r$ approximation). *Suppose we are given a matrix $M$ of rank $n$ whose singular value decomposition is*

$$M = \sum_{i=1}^{n} \sigma_i \mathbf{u}_i \mathbf{v}_i^\top \,,$$

*where the singular-values $(\sigma_i)$ are sorted in descending order. Then the best rank-$r$ approximation of $M$ for $r \le n$ is*

$$\mathcal{C}_r(M) = (\sum_{i=1}^{r} \sigma_i \mathbf{u}_i \mathbf{v}_t^\top) Q \,,$$

*where $Q \in \mathbb{R}^{r \times r}$ is an orthogonal matrix, and further the quality of its approximation is bounded by*

$$\|M - \mathcal{C}_r(M)\|^2 = \left( 1 - \frac{\sum_{i=1}^{r} \sigma_i^2}{\sum_{i=1}^{n} \sigma_i^2} \right) \|M\|^2 \,.$$

Thus if we used Algorithm 2 with exact rank-$r$ approximation of the gradients, we would converge at rate dictated by the eigen-spectrum of the gradients. If the singular values are 'top-heavy' i.e. the largest $r$ values are significantly larger than the rest, then a rank-$r$ approximation is quite accurate. As demonstrated in (Wang et al., 2018), the eigen-spectrum of stochastic gradients in common deep learning tasks is indeed 'top-heavy'. Thus we can expect $\delta_{e,r}$ to be bounded away from 0 even for very small $r$ (e.g. 1 or 2). Of course computing the actual top eigenvectors of the stochastic gradients is very computationally expensive, and more-over is not linear (and hence does not support *reduce*).

## A.2 Subspace iteration

The key innovation in POWERSGD is to use only a *single* step of subspace (or power) iteration to give a fast low rank approximation (Stewart & Miller, 1975) to the given matrix, which in our case is a stochastic gradient. However, a single step of subspace iteration in general does not result in an adequate low-rank approximation of the input matrix. To combat this, and to at the same time reduce the variance of the stochastic gradient approximation compared to the full (deterministic) gradient, we propose the *reuse* of the low-rank approximation from the previous iteration as the starting point for the current iteration. This is in spite of the target matrices which are trying to approximate are *changing*, as the parameters evolve. Nevertheless, reuse here is justified because the full gradient does not change very fast (the gradient is Lipschitz by assumption) and we only perform a tiny update at each step, so can be assumed to be stationary within a small number of steps. Intuitively, by linearity of the subspace operation, the sequence of subspace steps with the reuse then is converging to the eigenvector of the averaged stochastic gradients over these steps, thus having a lower variance than the analogue without re-use, which has no such averaging effect.

For simplicity, we assume all matrices to be square and symmetric in this sub-section. These insights can be generalized to arbitrary matrices but with a substantial increase in complexity of exposition. Here, we simply note that for any non-square matrix $A$, we can instead consider

$$\tilde{A} = \begin{bmatrix} 0 & A \\ A^\top & 0 \end{bmatrix}$$

which is symmetric and has the same eigenvectors and eigenvalues as the original matrix $A$—see Stewart (1976) for more details on handling such cases.

We can now state an informal theorem about the convergence of subspace iteration.

**Theorem I.** *Suppose that we run subspace iteration as in* (3) *on a fixed matrix $A_t = M$. Also let $M = \sum_{i=1}^n \sigma_i \mathbf{u}_i \mathbf{u}_i^\top$ be the eigen decomposition of $M$ with $\sigma_1 \geq \ldots \sigma_r > \sigma_{r+1} \geq \cdots \geq \sigma_n$. Then there exists an orthogonal matrix $Q \in \mathbb{R}^{r \times r}$ such that*

$$\lim_{t=\infty} X_t = [\mathbf{u}_1, \ldots, \mathbf{u}_r]Q \,.$$

*In other words,* (3) *recovers the best rank-$r$ approximation of $M$ as long as there is a gap between the $\sigma_r$ and $\sigma_{r+1}$ eigenvalues.*

Suppose that at each iteration we receive a matrix $A_t \in \mathbb{R}^{n \times n}$ whose expectation is the same fixed matrix $M \in \mathbb{R}^{n \times n}$. Starting from an orthonormalized $X_0 \in \mathbb{R}^{n \times r}$ (i.e. $X_0^\top X_0 = I_r$), the rank-$r$ subspace iteration algorithm performs the following update:

$$X_{t+1} = \text{ORTHOGONALIZE}(A_t X_t) \,. \tag{3}$$

The final output of the algorithm (i.e.) the matrix approximation is $(A_{T+1} X_T) X_T^\top$. This closely resembles the method of POWERSGD as outlines in Algorithm 1. We recommend (Arbenz, 2016) for an in-depth analysis of the (non-varying) subspace iteration algorithm.

**Remark 2** (Orthogonalization is a linear operation). *We recall some more facts from linear algebra. For any square matrix $B$, there exists an orthogonal matrix $Q$ and a triangular matrix $R$ such that $QQ^\top = I$ and $B = QR$. This is true e.g. if we use Gram–Schmidt procedure to ortho-normalize $B$: Suppose* ORTHOGONALIZE$(B)$ *uses the Gram–Schmidt procedure to orthogonalize $B$. Then there exists a triangular matrix $R$ such that*

$$\text{ORTHOGONALIZE}(B) = BR^{-1} \,.$$

*Proof.* It is easy to see that for any orthogonal matrix $Q$, the matrix $[\mathbf{u}_1, \ldots, \mathbf{u}_r]Q$ is also orthogonal, and further is the fixed point of (3). In fact all rank-$r$ matrices which are fixed points of (3) are of this form.

We will use the observation in Remark 2 to rewrite the update (3) in a more convient fashion. There exist tringular matrices $R_0, \ldots, R_t$ such that

$$X_{t+1} = \text{ORTHOGONALIZE}(A_t X_t) = A_t X_t R_t^{-1} = (A_t A_{t-1} \cdots A_0) X_0 (R_0^{-1} R_1^{-1} \cdots R_t^{-1}) \,.$$

Thus $X_{t+1}$ can alternatively be written as

$$X_{t+1} = \text{ORTHOGONALIZE}((A_t A_{t-1} \cdots A_0) X_0) = \text{ORTHOGONALIZE}(M^{t+1} X_0) \,.$$

Here we assumed that the matrix was fixed i.e. $A_t = M$. Let us further assume that $X_0$ has a non-zero support on the first $r$ eigenvectors of $M$. Then, a gap in the eigenvalues $\sigma_r > \sigma_{r+1}$ implies that $\text{ORTHOGONALIZE}(M^{t+1} X_0)$ converges to $[\mathbf{u}_1, \ldots, \mathbf{u}_r]Q$. We refer to Chapter 7.2 of Arbenz (2016) for the actual proof of this fact. □

## A.3 Single/multi worker equivalence

The difference between the update as written in (2) and Algorithm 2 is that the error computation and compression is performed on the *aggregated* gradient $\mathbf{g}_t$ instead of on the individual workers' gradients $\mathbf{g}_{t,w}$. While in general these are not equivalent, the linearity of POWERSGD ensures that these are indeed equivalent. This implies that POWERSGD has the neat property that the algorithm is equivalent if run on $W$ workers or a single worker with a larger batch-size. This does not hold for most other schemes (e.g. sign based compression schemes, QSGD, etc.).

**Lemma 3** (Equivalence of single worker and multi worker updates). *The updates in* POWERSGD *(i.e. Algorithm 2 using Compressor 1) are equivalent to the updates* (2).

*Proof.* Consider the update performed by POWERSGD for abrtiary vectors $\{\mathbf{v}_w\}$. Let $\mathcal{C}(\mathbf{v}_w)$ be the compressed version of $\mathbf{v}_w$ for $w \in \{1, \ldots, W\}$. Then by design of POWERSGD, the following holds:

$$\text{DECOMPRESS}(\text{AGGREGATE}(\mathcal{C}(\mathbf{v}_1), \ldots, \mathcal{C}(\mathbf{v}_W))) = \text{DECOMPRESS}(\mathcal{C}(\frac{1}{W} \sum_w \mathbf{v}_w)) \,.$$

This implies that running the algorithm on multiple workers, or running it on a single worker with a larger batch-size is identical. In particular,

$$\text{DECOMPRESS}(\text{AGGREGATE}(\mathcal{C}(\mathbf{g}_{t,1} + \mathbf{e}_{t,1}), \ldots, \mathcal{C}(\mathbf{g}_{t,W} + \mathbf{e}_{t,W})))$$

$$= \text{DECOMPRESS}(\mathcal{C}(\frac{1}{W} \sum_w \mathbf{g}_{t,w} + \mathbf{e}_{t,w}))$$

$$= \text{DECOMPRESS}(\frac{1}{W} \mathcal{C}(\mathbf{g}_t + \mathbf{e}_t)) \,.$$

□

# B   Cluster specifications

- 8 nodes
- GPUs: 2× Nvidia GeForce GTX Titan X with 12 GB memory per node
- GPU connection: traversing PCIe and the SMP interconnect between NUMA nodes
- CPU: Intel Xeon E5-2680 v3 @ 2.50Ghz, 48 cores
- System memory: 251GiB
- Ethernet: 10Gbit/s SFI/SFP+
- *Fat tree* network topology
- Runing PYTORCH 1.1 on Anaconda Python 3.7

**Timings of collective communication operations**

The figure below shows timings for the NCCL backend, which is the default in our experiments, and the GLOO backend. Note that NCCL does not support the 'gather' operation in PYTORCHat the time of writing.

# C   Convergence curves

Figure 4: Convergence curves of POWERSGD with varying rank. This figure is meant to give context to the final results and timings presented in Table 3. In two different tasks, POWERSGD with high enough rank can achieve the test quality of full-precision SGD with lower wall-clock duration. Contrary to Table 3, these timings include testing overhead at the end of each epoch, checkpointing, and other bookkeeping. Shaded areas show the min—max values over 3 replications of the experiments.

Figure 5: Convergence curves comparing POWERSGD to the Signum optimizer Bernstein et al. (2019) (with tuned learning rate). Out of the compared methods, Signum came out as the most competitive. This figure is meant to give context to the final results and timings presented in Table 6. Contrary to Table 3, these timings include testing overhead at the end of each epoch, checkpointing, and other bookkeeping. Shaded areas show the min—max values over 3 replications of the experiments.

# D  Language Modeling with Transformers

In this case study, we assess PowerSGD's universality and ease of tuning. We implemented PowerSGD communication in Facebook AI Research's `fairseq` library (Ott et al., 2019). We trained fairseq's language modeling example[2] with transformers (Baevski & Auli, 2019) on Google's public cloud. The communication infrastructure, hardware, number of workers (32), and model architecture are all different from any experiments we have conducted before. See Table 8 for details.

The results of our experiments for various ranks are shown in Figure 6 and Table 9. For this task, we need a higher rank than previously (32 vs 4) to achieve a validation loss comptetitive to uncompressed SGD. We hypothesize this may be due differences in architecture to the cosine learning rate schedule. Nevertheless, even at this higher rank, we achieve a time-to-accuracy (to loss = 5) of around $1.5\times$ and a compression ratio of $14\times$. These numbers could probably be further improved by re-tuning learning-rate-related hyperparameters.

Table 8: Experimental setting for the experiments in Appendix D

| | |
|---|---|
| Dataset | WikiText-103 |
| Architecture | Transformer-based (Baevski & Auli, 2019) |
| Framework & defaults | `https://github.com/pytorch/fairseq/tree/920b85d4bd39e181229db5639c701c854c83ec5c/` `examples/language_model` |
| Number of workers | 32 |
| Backend | NCCL (fastest in PYTORCH) |
| Hardware | n1-standard-8 nodes on Google Cloud with 1 Nvidia Tesla K80 GPU |
| Hyperparameters | Taken from the example, not re-tuned, |
| | with minor changes for the higher number of workers and different GPU memory: |
| `lr period updates` | 16875 |
| `max update` | 17875 |
| `max tokens (valid)` | 1536 (to fit on a K80 gpu) |
| `tokens per sample` | 1536 (to fit on a K80 gpu) |
| `warmup updates` | 1000 |
| `update freq` | [1] — don't aggregate multiple mini-batches locally |
| Optimizer | original: Nesterov accelerated gradient, we just added PowerSGD for communication |
| Learning rate | original cosine schedule from the example |
| Float precision | 32-bit (16-bit is unavailable on the K80) |
| Repetitions | 1 |

Figure 6: Language Modeling on WIKITEXT-2 with Transformers. With a large enough rank, POWERSGD can roughly match the validation loss of full-precision SGD in the same number of iterations. A speedup of $1.5\times$ in time-to-accuracy (loss=5) is achieved with a rank of 16.

Table 9: POWERSGD for Language Modeling with Transformers. With rank 32, POWERSGD achieves similar validation loss to uncompressed SGD in the same number of update steps. At this rank, the compression ratio is $14\times$ and we can train the model in 12h compared to 20h for the baseline.

| Compression | Total training time<br>for 17875 updates | | Compression ratio | Validation loss<br>at 17875 updates |
|---|---|---|---|---|
| Uncompressed | | 20h | $1\times$ | 4.92 |
| Rank 4 | | 11h | $105\times$ | 5.58 |
| Rank 8 | | 11h | $55\times$ | 5.19 |
| Rank 16 | | 12h | $28\times$ | 5.03 |
| Rank 32 | | 13h | $14\times$ | 4.97 |
| | 4h  8h  12h 16h 20h | | | |

■ Forward pass   ■ Backward pass   ■ Gradient exchange including computation

# E  The need for error feedback

Figure 7: PowerSGD with and without error feedback compared. While rank-4 POWERSGD achieves the same test accuracy as full-precision SGD, the same method without error feedback does not converge to a good accuracy at all. Both experiments use the same learning rate that was tuned for full-precision SGD.

# F Network parameters

See Table 10 and Table 11 for an overview of parameters in the models used.

Table 10: Parameters in the ResNet18 architecture and their shapes. The table shows the per-tensor compression ratio achieved by rank-$r$ POWERSGD.

| Parameter | Gradient tensor shape | Matrix shape | Uncompressed | Compression |
|---|---|---|---|---|
| layer4.1.conv2 | $512 \times 512 \times 3 \times 3$ | $512 \times 4608$ | 9216 KB | $461/r \times$ |
| layer4.0.conv2 | $512 \times 512 \times 3 \times 3$ | $512 \times 4608$ | 9216 KB | $461/r \times$ |
| layer4.1.conv1 | $512 \times 512 \times 3 \times 3$ | $512 \times 4608$ | 9216 KB | $461/r \times$ |
| layer4.0.conv1 | $512 \times 256 \times 3 \times 3$ | $512 \times 2304$ | 4608 KB | $419/r \times$ |
| layer3.1.conv2 | $256 \times 256 \times 3 \times 3$ | $256 \times 2304$ | 2304 KB | $230/r \times$ |
| layer3.1.conv1 | $256 \times 256 \times 3 \times 3$ | $256 \times 2304$ | 2304 KB | $230/r \times$ |
| layer3.0.conv2 | $256 \times 256 \times 3 \times 3$ | $256 \times 2304$ | 2304 KB | $230/r \times$ |
| layer3.0.conv1 | $256 \times 128 \times 3 \times 3$ | $256 \times 1152$ | 1152 KB | $209/r \times$ |
| layer2.1.conv2 | $128 \times 128 \times 3 \times 3$ | $128 \times 1152$ | 576 KB | $115/r \times$ |
| layer2.1.conv1 | $128 \times 128 \times 3 \times 3$ | $128 \times 1152$ | 576 KB | $115/r \times$ |
| layer2.0.conv2 | $128 \times 128 \times 3 \times 3$ | $128 \times 1152$ | 576 KB | $115/r \times$ |
| layer4.0.shortcut.0 | $512 \times 256 \times 1 \times 1$ | $512 \times 256$ | 512 KB | $171/r \times$ |
| layer2.0.conv1 | $128 \times 64 \times 3 \times 3$ | $128 \times 576$ | 288 KB | $105/r \times$ |
| layer1.1.conv1 | $64 \times 64 \times 3 \times 3$ | $64 \times 576$ | 144 KB | $58/r \times$ |
| layer1.1.conv2 | $64 \times 64 \times 3 \times 3$ | $64 \times 576$ | 144 KB | $58/r \times$ |
| layer1.0.conv2 | $64 \times 64 \times 3 \times 3$ | $64 \times 576$ | 144 KB | $58/r \times$ |
| layer1.0.conv1 | $64 \times 64 \times 3 \times 3$ | $64 \times 576$ | 144 KB | $58/r \times$ |
| layer3.0.shortcut.0 | $256 \times 128 \times 1 \times 1$ | $256 \times 128$ | 128 KB | $85/r \times$ |
| layer2.0.shortcut.0 | $128 \times 64 \times 1 \times 1$ | $128 \times 64$ | 32 KB | $43/r \times$ |
| linear | $10 \times 512$ | $10 \times 512$ | 20 KB | $10/r \times$ |
| conv1 | $64 \times 3 \times 3 \times 3$ | $64 \times 27$ | 7 KB | $19/r \times$ |
| Bias vectors (total) | | | 38 KB | None |
| **Total** | | | 43 MB | $243/r \times$ |

Table 11: Parameters in the LSTM architecture and their shapes. The table shows the per-tensor compression ratio achieved by rank-$r$ POWERSGD.

| Parameter | Gradient tensor shape | Matrix shape | Uncompressed | Compression |
|---|---|---|---|---|
| encoder | $28869 \times 650$ | $28869 \times 650$ | 73300 KB | $636/r \times$ |
| rnn-ih-l0 | $2600 \times 650$ | $2600 \times 650$ | 6602 KB | $520/r \times$ |
| rnn-hh-l0 | $2600 \times 650$ | $2600 \times 650$ | 6602 KB | $520/r \times$ |
| rnn-ih-l1 | $2600 \times 650$ | $2600 \times 650$ | 6602 KB | $520/r \times$ |
| rnn-hh-l1 | $2600 \times 650$ | $2600 \times 650$ | 6602 KB | $520/r \times$ |
| rnn-ih-l2 | $2600 \times 650$ | $2600 \times 650$ | 6602 KB | $520/r \times$ |
| rnn-hh-l2 | $2600 \times 650$ | $2600 \times 650$ | 6602 KB | $520/r \times$ |
| Bias vectors (total) | | | 174 KB | None |
| **Total** | | | 110 MB | $310/r \times$ |

# G Compressor implementation details

## G.1 Random Block

This implements compression for error-feedback with momentum (Algorithm 2).

---
**Algorithm 3** Random Block compression

---
1: **function** COMPRESS(update matrix $M \in \mathbb{R}^{n \times m}$)
2:     Treat $M$ as a vector of length $nm$.
3:     Sample an index $s$ uniformly between 0 and $nm - 1$, using the same seed on all workers.
4:     The block length $b$ is set to $(m + n)r$ to match rank-$r$ POWERSGD.
5:     **return** A consequtive memory slice $S = M(s : s + b)$.
6: **end function**
7: **function** AGGREGATE+DECOMPRESS(worker's slices $S_1 \dots S_W$)
8:     $\hat{M} \leftarrow \mathbf{0} \in \mathbb{R}^{n \times m}$
9:     $\hat{M}(s : s + b) \leftarrow \frac{1}{W} \sum_{i=1}^{W} S_i$           ▷ using all-reduce
10:     **return** $\hat{M}$
11: **end function**

---

## G.2 Random K

This implements compression for error-feedback with momentum (Algorithm 2).

---
**Algorithm 4** Random $K$ compression

---
1: **function** COMPRESS(update matrix $M \in \mathbb{R}^{n \times m}$)
2:     Treat $M$ as a vector of length $nm$.
3:     The number of samples $b$ is set to $(m + n)r$ to match rank-$r$ POWERSGD.
4:     Sample a set of $b$ indices $I$ without replacement, using the same seed on all workers.
5:     **return** Looked up values $S = M(I)$.
6: **end function**
7: **function** AGGREGATE+DECOMPRESS(worker's values $S_1 \dots S_W$)
8:     $\hat{M} \leftarrow \mathbf{0} \in \mathbb{R}^{n \times m}$
9:     $\hat{M}(I) \leftarrow \frac{1}{W} \sum_{i=1}^{W} S_i$           ▷ using all-reduce
10:     **return** $\hat{M}$
11: **end function**

---

**Sampling of indices** We sample random indices on the CPU using Numpy. This operation is relatively expensive. Together with the many random lookups, this explains why Random $K$ compression is significantly slower than Random Block compression.

## G.3 Sign+Norm

This implements compression for error-feedback with momentum (Algorithm 2).

---
**Algorithm 5** Sign+Norm compression

---
1: **function** COMPRESS(update matrix $M \in \mathbb{R}^{n \times m}$)
2:     Compute the signs $S \in \{-1, 1\}^{n \times m}$ of $M$
3:     Compute the $L_1$ norm $\ell$ of $M$.
4:     **return** $(\ell, S)$
5: **end function**
6: **function** AGGREGATE+DECOMPRESS(worker's norms $\ell_1 \dots \ell_W$ and signs $S_1 \dots S_W$)
7:     **return** $\frac{1}{W} \sum_{i=1}^{W} \frac{\ell_i}{nm} S_i$       ▷ Executed on all workers using NCCL's all-gather
8: **end function**

---

Because PYTORCH does not natively support data types smaller than 8 bits per scalar, we use a C++ extension (Bernstein et al., 2019) to actually send single bits to other workers. The employed all-gather operation from NCCL is faster than aggregation using a parameter server using GLOO. We cannot implement a parameter server in NCCL due to lack of a 'gather' operation.

### G.4 Top K

This implements compression for error-feedback with momentum (Algorithm 2).

---
**Algorithm 6** Top $K$ compression

---
 1: **function** COMPRESS(update matrix $M \in \mathbb{R}^{n \times m}$)
 2:     Treat $M$ as a vector of length $nm$.
 3:     The number of samples $b$ is set to $(m + n)r$ to match rank-$r$ POWERSGD.
 4:     Construct a list of $b$ indices $I$ corresponding to the top absolute values in $M$.
 5:     **return** Looked up values $S = M(I)$ and indices $I$.
 6: **end function**
 7: **function** AGGREGATE+DECOMPRESS(worker's values $S_1 \dots S_W$ and indices $I_1 \dots I_W$)
 8:     $\hat{M} \leftarrow \mathbf{0} \in \mathbb{R}^{n \times m}$
 9:     **for** worker index $i$ in $1, \dots, W$ **do**
10:         $\hat{M}(I_i) \leftarrow \frac{1}{W} S_i$                          ▷ using all-gather in NCCL
11:     **end for**
12:     **return** $\hat{M}$
13: **end function**

---

The employed all-gather operation from NCCL is faster than aggregation using a parameter server using GLOO. We cannot implement a parameter server in NCCL due to lack of a 'gather' operation.

### G.5 Signum

This is our implementation of the Signum compression algorithm by Bernstein et al. (2019). We run it in its original form, without error feedback, with momentum of 0.9, and a learning rate tuned based on 5 experiments in the 16-worker setting.

---
**Algorithm 7** Signum compression

---
 1: **function** COMPRESS(update matrix $M \in \mathbb{R}^{n \times m}$)
 2:     Compute the signs $S \in \{-1, 1\}^{n \times m}$ of $M$
 3:     **return** $S$
 4: **end function**
 5: **function** AGGREGATE+DECOMPRESS(worker's signs $S_1 \dots S_W$)
 6:     **return** SIGN($\sum_{i=1}^{W} S_i$)              ▷ Majority vote, on all workers using NCCL's all-gather
 7: **end function**

---

Because PYTORCH does not natively support data types smaller than 8 bits per number, we use a C++ extension Zhao (2019) to actually send single bits to other workers. The employed all-gather operation from NCCL is faster than aggregation using a parameter server using GLOO. We cannot implement a parameter server in NCCL due to lack of a 'gather' operation.

### G.6 Atomo

This is our implementation of the Spectral Atomo algorithm presented by Wang et al. (2018). We run it in its original form, without error feedback, with momentum of 0.9, and a learning rate tuned based on 4 experiments in the 16-worker setting.

**Matix shape**   Atomo differs from POWERSGD in how it treats tensors as matrices. This results in lower compression at the same rank.

**Number of sampled components**  Atomo decomposes gradient matrices $M$ using a Singular Value Decomposition into $M \sim \sum_i U_{i:} S_{ii} V_{i:}^\top$ and importance-samples components from this summation based on probabilities derived from the absolute singular values $S_{ii}$. The probabilities are such, that the expected number of samples components is equal to the target rank $r$, but there is no guarantee. We modify the algorithm to always use exactly $r$ components, to allow for faster communication. We achieve this by repeating the sampling procedure until the number of selected components is $r$. This has no significant impact on the runtime performance.

---

**Algorithm 8** Rank-$r$ Spectral-Atomo compression

---

1: **function** COMPRESS(update matrix $M \in \mathbb{R}^{n \times m}$)
2:   $U, S, V \leftarrow \text{SVD}(M)$.                                    ▷ on CPU using Numpy, faster than PYTORCH
3:   Compute Atomo probabilities $p_1 \ldots p_k$ from $S_{11}, \ldots S_{kk}$.          ▷ see Wang et al. (2018).
4:   Sampling: include index $i$ independently with probability $p_i$.
5:   Repeat sampling until a set of $r$ indices $C$ is selected.          ▷ our modification (see above)
6:   **return** $\{(U_{i:} \cdot S_{ii}/p_i, V_{i:}) \mid i \in C\}$ as two matrices $U' \in \mathbb{R}^{n \times r}$ and $V' \in \mathbb{R}^{m \times r}$.
7: **end function**
8: **function** AGGREGATE+DECOMPRESS(rank-$r$ approximations $(U_1', V_1') \ldots (U_W', V_W')$ for each worker)
9:   **return** $\sum_{i=1}^{W} U_i' V_i'^\top$                                    ▷ using all-gather in NCCL
10: **end function**

---

The employed all-gather operation from NCCL is faster than aggregation using a parameter server using GLOO. We cannot implement a parameter server in NCCL due to lack of a 'gather' operation.

### G.7  Best-approximation POWERSGD

This variant is the same as POWERSGD (Algorithm 1), but with more steps of subspace iteration, and without reuse of previous steps. We find that 4 steps of subspace iterations (8 matrix multiplications) is enough to converge to the best low-rank approximation of gradient matrices, when measuring final test accuracy achieved by POWERSGD.

# H   Performance optimizations

Because we compare timings, we have aimed to optimize all compared optimizers to a similar level. For sign-based methods, we used a publicly available C++ library by Bernstein et al. (2019) to efficiently pack the signs into bitmaps, an operation which is not supported by PYTORCH natively. For Atomo, we have benchmarked the SVD operation on the GPU and CPU, and chose the faster CPU implementation. For all methods, we pack all gradient tensors into one flat buffer to reduce the number of communications. Where possible, we overlay communication with computation. Algorithms that do not support all-reduce are implemented using NCCL's all-gather, which is faster than a parameter server with GLOO.[3]

# I   Learning rate tuning

For each task and each optimization algorithm without error feedback, learning rates were tuned separately. For algorithms based on error feedback with momentum, we use the learning rate tuned for SGD.

Learning rates are defined as rates for 1 worker, and scaled linearly with 5-epoch warmup to the number of workers (16 by default). We tune them in the 16-worker setting.

We determine the best learning rate by comparing test accuracy of one replication after running the full number of epochs. We start training with 3 different learning rates, a factor 2 apart, based on commonly used rates for the optimizer, and if the best learning rate is either the lower or higher end, we extended the range.

For CIFAR10, the rates considered for SGD were [0.05, 0.1, 0.2], we chose 0.1. For rank-2 Spectral Atomo, we considered [0.025, 0.05, 0.1, 0.2] and chose 0.1. For Signum, we considered [2e-5, 5e-5, 1e-4, 2e-4] and chose 5e-5.

For WIKITEXT-2, the rates considered for SGD were [0.6, 1.25, 2.5, 5, 10], we chose 1.25. For Signum, we considered [2e-4, 1e-1, 5e-5, 1e-5, 1e-6], and chose 1e-5.

We have not tuned the momentum parameter or $L_2$, weight decay parameters or learning rate schedule for any experiment.