[Reviews · NeurIPS 2019]

Reviewer 1



Update: I have carefully read the authors' rebuttal. I have raised by score to 6 from 5 to reflect their clarification about Figure 3 and Table 6. It still seems that the speedups of the current formulation are often not of great practical significance, except for the language model which was able to give 2x wall clock speedup. As another reviewer noted, it is disappointing that the overall training time is not reported in the main paper, instead of the average batch time, as that makes it unclear whether latency times and other overheads between batches might be a significant concern. The author rebuttal notes that Appendix C shows time-to-accuracy, which would be good to mention in the main paper. But those results still appear mixed: for CIFAR10 SGD beats Rank1 and seems only competitive with Ranks 2,4, whereas for Language model all Ranks seem to convincingly beat SGD. So it's not often this approach in current form will produce significant improvements for a given new task. Overall this approach appears promising but lacks any significant theoretical support. Furthermore, the experiments offered mainly provide an existence proof that some wall-clock speed up is achieved using gradient compression, without putting the these results in a significant context (e.g. what are limitations, such as when the single step approximation of the power iteration fails and gives worse test accuracy?). PowerSGD reduces gradient communication significantly in some experiments, but the net speedup is not proportional (e.g.for LSTM experiment, 10x reduction only yields 2x speedup). However, even that claimed improvement seems misleading. Figure 3 shows that using modern NCCL, nearly 8x speedup on 16 GPUS across 8 machines is achieved by not only (Rank-2) PowerSGD, but also SGD and Signum. Thus, tables 6 and 7 seem misleading, because although they seem to indicate significant practical speedups (e.g. 134 ms / epoch for PowerSGD versus 300ms for SGD for language model LSTM), that actually seems to be the case only when using the slow GLOO backend, where SGD and PowerSGD have significant differences as shown in Figure 3, and not when the modern, commonly available, NCCL is used for all-reduce. If that is the case, then this approach is not clearly of much practical significant as is for modern practiioners (who typically do use NCCL). So, the significance of the contributions of this work are unclear, especially given the lack of theoretical justifications and limited scope of the experimental work. Algorithm 1 (rank-r PowerSGD compression) is presented without justification. It is not clear from the experiments when this works well or fails, or what existing work this is based on (there are no citations offered).

Reviewer 2



This paper studies gradient compression methods for distributed optimization. The existing sign-based approach and top-k method relies on all-gather operation, which does not scale well with increasing number of workers. In this paper, a rank-r powerSGD compression method is proposed that allows using all-reduce, which has better scalability than all-gather. The rank-r powerSGD compression utilizes the one-step power iteration and is therefore computationally inexpensive. Extensive experiments are conducted to demonstrate the efficiency of the proposed method. It is good to see the ablation study in the experiment. The paper is well written. Overall, I this is a good paper, but the theoretical parts and experiments can be further strengthened. It is not clear to me why powerSGD enjoys linearity. In particular, why does Lemma 3 in supplementary material hold? As can be seen in Algorithm 1, the construction of matrix Q involves matrix multiplication between M_i and M_j for i \not= j, and therefore \hat{P}Q^T \not= 1/W\sum_{i=1}^W\hat{P}_iQ_i^T, where \hat{P}_i and Q_i are obtained by performing compression on each i-th worker's gradient. Due to this, I do not see why powerSGD can have linearity. There is no rigorous convergence analysis of Algorithm 2 and Algorithm 1. Only a short of description of what techniques can be applied in the supplementary material. A similar setting is considered in [Yu et al., 2018], which proposed a PCA-based compression for all-reduce. It would be good to compare the proposed method with it. In Section 4.1, an unbiased low-rank approximation without feedback is compared. Why don't we just compare to powerSGD without feedback? The testing accuracy 94.3% of ResNet18 trained with SGD on CIFAR-10 is much better than the 91.25% result in original paper [He et al., 2016]. I wonder which difference in the experimental setting contributes to such improvement? Distributed training on a small scale dataset such as CIFAR-10 is not very interesting. It is more convincing to conduct experiment on training a deep ResNet on the ImageNet dataset. Yu et al. GradiVeQ: Vector Quantization for Bandwidth-Efficient Gradient Aggregation in Distributed CNN Training. NIPS 2018. He et al. Deep Residual Learning for Image Recognition. CVPR 2016. ===================================== I have read the rebuttal, and my score remains the same. I would encourage the authors to take time to study the convergence with k-step power method, and perform the experiments on larger dataset such as ImageNet.

Reviewer 3



Updated review: After carefully reading the rebuttal, part of my concerns are addressed as the authors claim they will extend the experimental study to a larger scale. However, I don't wish to change my overall score. I agree with Reviewer 1 and Reviewer 3 that providing convergence analysis will strengthen this work, but it may take time. Moreover, I insist on my suggestion that i) the authors should consider studying PowerSGD under the setting that global batch size is fixed since this will lead to more promising speedups. ii) end-to-end speedups need to be clearly illustrated e.g. adding one table that indicates speedups of PowerSGD comparing to various baselines on converging to certain accuracies. ------------------------------------------------------------------------------------------------- This paper proposed PowerSGD, a low-rank based gradient compression algorithm for speeding up distributed training. Compared to the previously proposed method ATOMO [1], PowerSGD avoids the computationally expensive SVD step in ATOMO to attain better scalability. PowerSGD is a biased gradient compression technique, which makes it hard to conduct solid theoretical analysis. Moreover, PowerSGD is the first gradient compression algorithm that is compatible with all-reduce based distributed applications compared to most of the previously proposed communication efficient distributed training algorithms. The paper is very well written and well motivated. Extensive experimental results indicate that PowerSGD has good scalability and end-to-end convergence performance. [1] https://papers.nips.cc/paper/8191-atomo-communication-efficient-learning-via-atomic-sparsification

[Author Response · NeurIPS 2019]

We thank the reviewers for their insightful comments and encouraging feedback. We hope that the concerns raised are addressed adequately below and that our work will be appropriately re-evaluated.

**Speedups (R1)** Reviewer 1 raises two concerns about speedups which we believe to be based on a misunderstanding. Firstly, our large reductions in communication (e.g. $10\times$) lead to smaller reductions in wall-clock time (e.g. $2\times$). We think this is expected, as all mentioned wall-clock times *include forward and backward passes* in addition to gradient compression and communication. A $2\times$ reduction in this metric seems significant. We will clarify this in the paper.

Secondly, the reviewer suspects that Tables 6 and 7 show timings for the slower GLOO backend. Let us clarify that all such timings are measured in default conditions: NCCL, all-reduce, 16 GPUs, and end-to-end as described before. The scaling plots in Figure 3 show speedups of 9.3 (PowerSGD) vs 7.1 (SGD) on Cifar over single worker SGD. This is consistent with the 23% savings (9.3 vs. 7.1) reported in Table 6. We include results for an LSTM (Table 7) for completeness. The LSTM's speedups are better due to their higher communication-to-computation ratio.

**Failure cases for PowerSGD (R1)** We agree with Reviewer 1 that an outline of when PowerSGD works and when it breaks would be helpful. To date, we have not observed any failure cases of the 1-step power iteration in the algorithm. To achieve good accuracy in the same number of steps as SGD, a sufficiently high rank (2 or 4 in practice) is required.

**Larger models and clusters (R1, R3, R5)** We are currently running additional experiments on a larger cluster (64 GPUs) with larger models (ResNet-50). This should further test the effect of network latency (R5). Extrapolating the scaling plots in Figure 3, we expect PowerSGD to perform favorably in those conditions.

**Convergence of Algorithms 1 and 2 (R1, R3)** While we do not currently include an end-to-end convergence proof for PowerSGD, each of its core components are well studied. Algorithm 2 (EF-SGD with Momentum) adds momentum to the well-studied EF-SGD algorithm (as in Karimireddy et al. 2019). EF-SGD is guaranteed to converge if the compressor $\mathcal{C}$ satisfies $\|X - \mathcal{C}(X)\|_2^2 \leq (1 - \delta)\|X\|_2^2$. This condition is satisfied by PowerSGD with SVD for best rank-$k$ approximation (see Appendices A.1 and A.2). The cheaper 1-step power iteration with warm start is akin to the famous Oja's algorithm (Oja, 1982) and is empirically shown to yield the same performance as a full SVD.

**Linearity of PowerSGD (R3)** We use the term linearity to mean that PowerSGD on a single worker with gradient matrix $M := (M_1 + M_2)/2$ is equivalent to PowerSGD with two workers with their own gradients $M_1$ and $M_2$ (for any number of workers and any split of $M$.) To see that this holds, consider that $P$ in line 4 of Algorithm 1 in the two-worker example amounts to $P = \frac{1}{2}(M_1 + M_2)Q = \bar{M}Q$. The matrices $M_1$ and $M_2$ are never multiplied with each other. The same is true for $Q$ in line 7. This makes PowerSGD just a function of the average gradient $\bar{M}$.

**PowerSGD without feedback (R3)** Because PowerSGD's very-low-rank gradient approximations are coarse, it required error feedback to converge in our experiments. We will include the requested comparison in the Appendix.

**GradiVeQ (Yu et al. 2018) (R3)** We thank Reviewer 3 for pointing us to this interesting work. We will include this method in our discussion.

**High Cifar-10 accuracy (R3)** We use a ResNet-18 based on `torchvision`. Compared to the ResNet-20 model used for Cifar-10 in the original paper, the layers have more feature maps (are wider), explaining the superior performance. He et al. use this wider architecture for ImageNet.

**Global batch size (R5)** Reviewer 5 mentions that some related papers scale the number of workers while keeping the global batch size fixed. Most of the work we are aware of instead keep the local batch size fixed (e.g. Goyal et al. (2017)) since it better utilizes the computational power of the workers. Moreover, if we kept a fixed global batch size the computation performed per bit communicated would decrease with the number of workers—only further favoring compressed algorithms such as ours.

**End-to-end speedup results (R5)** Time-to-accuracy results, as requested by Reviewer 5, can currently be found in Appendix C of the submission. We will consider including these plots in the main paper.

Goyal, P., et al. "Accurate, large minibatch SGD: Training ImageNet in 1 hour." arXiv 2017.
He, K., et al. "Deep residual learning for image recognition." CVPR 2016.
Karimireddy, S.P. et al., "Error feedback fixes SignSGD and other gradient compression schemes." ICML 2019.
Oja, E. "Simplified neuron model as a principal component analyzer." Journal of Mathematical Biology, 1982.


[Meta-Review · NeurIPS 2019]

The authors propose a very interesting technique that is all-reduce compatible, for communication efficient learning. During the rebuttal phase the authors addressed most of the comments raised by the reviewers. The authors are strongly encouraged to address before the camera ready: - reporting end-to-end speedup of PowerSGD - adding a comparison of speedup curves comparing to various baselines on converging to certain accuracies.